# WOULDA, COULDA, SHOULDA:
# COUNTERFACTUALLY-GUIDED POLICY SEARCH

**Lars Buesing, Théophane Weber, Yori Zwols,**
**Sébastien Racanière, Arthur Guez, Jean-Baptiste Lespiau, Nicolas Heess**
DeepMind
lbuesing@google.com

## ABSTRACT

Learning policies on data synthesized by models can in principle quench the thirst of reinforcement learning algorithms for large amounts of real experience, which is often costly to acquire. However, simulating plausible experience de novo is a hard problem for many complex environments, often resulting in biases for model-based policy evaluation and search. Instead of de novo synthesis of data, here we assume logged, real experience and model alternative outcomes of this experience under *counterfactual* actions, i.e. actions that were not actually taken. Based on this, we propose the Counterfactually-Guided Policy Search (CF-GPS) algorithm for learning policies in POMDPs from off-policy experience. It leverages structural causal models for counterfactual evaluation of arbitrary policies on individual off-policy episodes. CF-GPS can improve on vanilla model-based RL algorithms by making use of available logged data to de-bias model predictions. In contrast to off-policy algorithms based on Importance Sampling which re-weight data, CF-GPS leverages a model to explicitly consider alternative outcomes, allowing the algorithm to make better use of experience data. We find empirically that these advantages translate into improved policy evaluation and search results on a non-trivial grid-world task. Finally, we show that CF-GPS generalizes the previously proposed Guided Policy Search and that reparameterization-based algorithms such Stochastic Value Gradient can be interpreted as counterfactual methods.

## 1 INTRODUCTION

Imagine that a month ago Alice had two job offers from companies $a_1$ and $a_2$. She decided to join $a_1$ because of the larger salary, in spite of an awkward feeling during the job interview. Since then she learned a lot about $a_1$ and recently received information about $a_2$ from a friend, prodding her now to imagine what would have happened had she joined $a_2$. Re-evaluating her decision in hindsight in this way, she concludes that she made a regrettable decision. She could and should have known that $a_2$ was a better choice, had she only interpreted the cues during the interview correctly... This example tries to illustrate the everyday human capacity to reason about alternate, counterfactual outcomes of past experience with the goal of "mining worlds that could have been" (Pearl & Mackenzie, 2018). Social psychologists theorize that such cognitive processes are beneficial for improving future decision making (Roese, 1997). In this paper we aim to leverage possible advantages of counterfactual reasoning for learning decision making in the reinforcement learning (RL) framework.

In spite of recent success, learning policies with standard, model-free RL algorithms can be notoriously data inefficient. This issue can in principle be addressed by learning policies on data synthesized from a model. However, a mismatch between the model and the true environment, often unavoidable in practice, can cause this approach to fail (Talvitie, 2014), resulting in policies that do not generalize to the real environment (Jiang et al., 2015). Motivated by the introductory example, we propose the Counterfactually-Guided Policy Search (CF-GPS) algorithm: Instead of relying on data synthesized *from scratch* by a model, policies are trained on model predictions of alternate outcomes of past experience from the true environment under *counterfactual* actions, i.e. actions that had not actually been taken, while everything else remaining the same (Pearl, 2009). At the heart

of CF-GPS are structural causal models (SCMs) which model the environment with two ingredients (Wright, 1920): 1) Independent random variables, called *scenarios* here, summarize all aspects of the environment that cannot be influenced by the agent, e.g. the properties of the companies in Alice's job search example. 2) Deterministic transition functions (also called *causal mechanisms*) take these scenarios, together with the agent's actions, as input and produce the predicted outcome. The central idea of CF-GPS is that, instead of running an agent on scenarios sampled de novo from a model, scenarios are inferred in hindsight from given off-policy data, and then evaluate and improve the agent on these *specific* scenarios using given or learned causal mechanisms (Balke & Pearl, 1994). CF-GPS can be regarded as a meta-algorithm that extends a given model-base RL algorithm by "grounding" or "anchoring" model-based predictions in inferred scenarios. As a result, this approach explicitly allows to trade-off historical data for model bias. We show empirically in a conceptually simple setting, where unknown initial states are inferred in hindsight and re-used to evalute to counterfactual actions, that this can mitigate model mismatch. CF-GPS differs substantially from standard off-policy RL algorithms based on Importance Sampling (IS), where historical data is *re-weighted* with respect to the importance weights to evaluate or learn new policies (Precup, 2000). In contrast, CF-GPS explicitly reasons counterfactually about given off-policy data. Our main contributions are:

1. We formulate model-based RL in POMDPs in terms of structural causal models, thereby connecting concepts from reinforcement learning and causal inference.

2. We provide the first results, to the best of our knowledge, showing that counterfactual reasoning in structural causal models on off-policy data can facilitate solving non-trivial RL tasks.

3. We show that two previously proposed classes of RL algorithms, namely Guided Policy Search (Levine & Koltun, 2013) and Stochastic Value Gradient methods (Heess et al., 2015), can be interpreted as counterfactual methods, opening up possible generalizations.

The paper is structured as follows. We first give a self-contained, high-level recapitulation of structural causal models and counterfactual inference, as these are less widely known in the RL and generative model communities. In particular we show how to model POMDPs with SCMs. Based on this exposition, we first consider the task of policy evaluation and discuss how we can leverage counterfactual inference in SCMs to improve over standard model-based methods. We then generalize this approach to the policy search setting resulting in the CF-GPS algorithm. We close by highlighting connections to previously proposed algorithms and by discussing assumptions and limitations of the proposed method.

## 2 PRELIMINARIES

We denote random variables (RVs) with capital letters, e.g. $X$, and particular values with lower caps, e.g. $x$. For a distribution $P$ over a vector-valued random variable $X$, we denote the marginal over $Y \subset X$ by $P_Y$ (and density $p_Y$); however we often omit the subscript if it is clear from the context, e.g. as in $Y \sim P$. We assume the episodic, Partially Observable Markov Decision Process (POMDP) setting with states $S_t$, actions $A_t$ and observations $O_t$, for $t = 1, \ldots, T$. For ease of notation, we assume that $O_t$ includes the reward $R_t$. The undiscounted return is denoted by $G = \sum_{t=1}^{T} R_t$. We consider stochastic policies $\pi(a_t|h_t)$ over actions conditioned on observation histories $H_t = (O_1, A_1, \ldots, A_{t-1}, O_t)$. We denote the resulting distribution over trajectories $\mathcal{T} = (S_1, O_1, A_1, \ldots, A_{T-1}, S_T, O_T)$ induced by running $\pi$ in the environment with $\mathcal{T} \sim \mathfrak{P}^\pi$ and the corresponding density by $\mathfrak{p}^\pi(\tau)$.

### 2.1 STRUCTURAL CAUSAL MODELS

**Definition 1** (Structural causal model). *A structural causal model (SCM) $\mathcal{M}$ over $X = (X_1, \ldots, X_N)$ is given by a DAG $\mathcal{G}$ over nodes $X$, independent noise RVs $U = (U_1, \ldots, U_N)$ with distributions $P_{U_i}$ and functions $f_1, \ldots, f_N$ such that $X_i = f_i(\mathrm{pa}_i, U_i)$, where $\mathrm{pa}_i \subset X$ are the parents of $X_i$ in G. An SCM entails a distribution $P$ with density $p$ over $(X, U)$.*

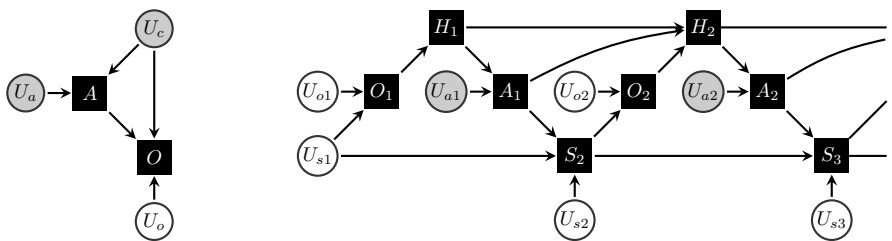

Figure 1: Structural causal models (SCMs) model environments using random variables $U$ (circles, 'scenarios'), that summarize immutable aspects, some of which are observed (grey), some not (white). These are fed into deterministic functions $f_i$ (black squares) that approximate causal mechanisms. **Left:** SCM for a contextual bandit with context $U_c$, action $A$, feedback $O$ and scenario $U_o$. **Right:** SCM for a POMDP, with initial state $U_{s1} = S_1$, states $S_t$ and histories $H_t$. The mechanism that generates the actions $A_t$ is the policy $\pi$.

We also refer to $U$ as scenarios and to $f_i$ as causal mechanisms. We give a (broad) definition of an *intervention* in an SCM. This also includes what is known as stochastic interventions or mechanism changes (Korb et al., 2004) which generalize atomic interventions (Pearl, 2009).

**Definition 2** (Intervention in SCM). *An intervention $I$ in an SCM $\mathcal{M}$ consists of replacing some of the original $f_i(\mathrm{pa}_i, U_i)$ with other functions $f_i^I(\mathrm{pa}_i^I, U_i)$ where $\mathrm{pa}_i^I$ are the parents in a new DAG $\mathcal{G}^I$. We denote the resulting SCM with $\mathcal{M}^{\mathrm{do}(I)}$ with distribution $P^{\mathrm{do}(I)}$ and density $p^{\mathrm{do}(I)}$.*

**SCM representation of POMDPs**  We can represent any given POMDP (under a policy $\pi$) by an SCM $\mathcal{M}$ over trajectories $\mathcal{T}$ in the following way. We express all conditional distributions, e.g. the transition kernel $P_{S_{t+1}|S_t, A_t}$, as deterministic functions with independent noise variables $U$, such as $S_{t+1} = f_{st}(S_t, A_t, U_{st})$. This is always possible using auto-regressive uniformization, see Lemma 2 in the appendix. The DAG $\mathcal{G}$ of the resulting SCM is shown in fig. 1. This procedure is closely related to the 'reparameterization trick' for models with location-scale distributions (Kingma & Welling, 2013; Rezende et al., 2014). We denote the distribution over $\mathcal{T}$ entailed by the SCM with $P^\pi$ and its density by $p^\pi$ to highlight the role of $\pi$; note the difference to the true environment distribution $\mathfrak{P}^\pi$ with density $\mathfrak{p}^\pi$. Running a different policy $\mu$ instead of $\pi$ in the environment can be expressed as an intervention $I(\pi \to \mu)$ consisting of replacing $A_t = f_\pi(H_t, U_{at})$ by $A_t = f_\mu(H_t, U_{at})$. We denote the resulting model distribution over trajectories with $P^{\mathrm{do}(I(\pi \to \mu))} = P^\mu$ (analogously $p^\mu$).

**Intuition**  Here, we illustrate the main advantage of SCMs using the example of Alice's job choice from the introduction. We model it as contextual bandit with feedback shown in fig. 1. Alice has some initial knowledge given by the context $U_c$ that is available to her before taking action $A$ of joining company $A = a_1$ or $A = a_2$. We model Alice's decision as $A = f_\pi(U_c, U_a)$, where $U_a$ captures potential indeterminacy in Alice's decision making. The outcome $O = f_o(A, U_c, U_o)$ also depends on the scenario $U_o$, capturing all relevant, unobserved and highly complex properties of the two companies such as working conditions etc. Given this model, we can reason about alternate outcomes $f_o(a_1, u_c, u_o)$ and $f_o(a_2, u_c, u_o)$ for *same* the scenario $u_o$. This is not possible if we only model the outcome on the level of the conditional distribution $P_{O|A, U_c}$.

## 2.2 COUNTERFACTUAL INFERENCE IN SCMS

For an SCM over $X$, we define a *counterfactual query* as a triple $(\hat{x}_o, I, X_q)$ of observations $\hat{x}_o$ of some variables $X_o \subset X$, an intervention $I$ and query variables $X_q \subset X$. The semantics of the query are that, having observed $\hat{x}_o$, we want to infer what $X_q$ would have been had we done intervention $I$, while '*keeping everything else the same*'. Counterfactual inference (CFI) in SCMs answers the query in the following way (Balke & Pearl, 1994):

1. Infer the unobserved noise source $U$ conditioned on the observations $\hat{x}_o$, i.e. compute $p(U|\hat{x}_o)$ and replace the prior $p(U)$ with $p(U|\hat{x}_o)$. Denote the resulting SCM by $\mathcal{M}_{\hat{x}_o}$.

2. Perform intervention $I$ on $\mathcal{M}_{\hat{x}_o}$. This yields $\mathcal{M}_{\hat{x}_o}^{\mathrm{do}(I)}$, which entails the counterfactual distribution $p^{\mathrm{do}(I)|\hat{x}_o}(x)$. Return the marginal $p^{\mathrm{do}(I)|\hat{x}_o}(x_q)$.

Note that our definition explicitly allows for partial observations $X_o \subset X$ in accordance with Pearl (2009). A sampled-based version, denoted as CFI, is presented in Algorithm 1. An interesting property of the counterfactual distribution $p^{\mathrm{do}(I)|\hat{x}_o}$ is that marginalizing it over observations $\hat{x}_o$ yields an unbiased estimator of the density of $X_q$ under intervention $I$.

**Lemma 1** (CFI for simulation). *Let observations $\hat{x}_o \sim p$ come from a SCM $\mathcal{M}$ with density $p$. Then the counterfactual density $p^{\mathrm{do}(I)|\hat{x}_o}$ is an unbiased estimator of $p^{\mathrm{do}(I)}$, i.e.*

$$\mathbb{E}_{\hat{x}_o \sim p}[p^{\mathrm{do}(I)|\hat{x}_o}(x)] \quad = \quad p^{\mathrm{do}(I)}(x)$$

The proof is straightforward and outlined in the Appendix A. This lemma and the marginal independence of the $U_i$ leads to the following corollary; the proof is given in the appendix.

**Corollary 1** (Mixed counterfactual and prior simulation from an SCM). *Assume we have observations $\hat{x}_o \sim p$. We can simulate from $\mathcal{M}$, under any intervention $I$, i.e. obtain unbiased samples from $\mathcal{M}^{\mathrm{do}(I)}$, by first sampling values $u_{\mathrm{CF}}$ for an arbitrary subset $U_{\mathrm{CF}} \subset U$ from the posterior $p(u_{\mathrm{CF}}|\hat{x}_o)$ and the remaining $U_{\mathrm{Prior}} := U \backslash U_{\mathrm{CF}}$ from the prior $p(u_{\mathrm{Prior}})$, and then computing $X$ with noise $u = u_{\mathrm{CF}} \cup u_{\mathrm{Prior}}$.*

The corollary essentially states that we can sample from the model $\mathcal{M}^I$, by sampling some of the $U_i$ from the prior, and inferring the rest from data $\hat{x}_o$ (as long as the latter was also sampled from $\mathcal{M}$). We will make use of this later for running a POMDP model on scenarios $U_{st}$ inferred from data while randomizing the action noise $U_{at}$. We note that the noise variables $U_{\mathrm{CF}}$ from the posterior $P_{U_{\mathrm{CF}}|\hat{x}_o}$ are not independent anymore. Nevertheless, SCMs with non-independent noise distributions arising from counterfactual inference, denoted here by $\mathcal{M}_{\hat{x}_o}$, are commonly considered in the literature (Peters et al., 2017).

**Intuition** Returning to Alice's job example from the introduction, we give some intuition for counterfactual inference in SCMs. Given the concrete outcome $\hat{o}$, under observed context $\hat{u}_c$ and having joined company $\hat{a} = a_1$, Alice can try to infer the underlying scenario $u_o \sim p(u_o|a_1, \hat{u}_c, \hat{o})$ that she experiences; this includes factors such as work conditions etc. She can then reason counterfactually about the outcome had she joined the other company, which is given by $f_o(a_2, \hat{u}_c, u_o)$. This can in principle enable her to make better decisions in the future in similar scenarios by changing her policy $f_\pi(A, U_c, U_a)$ such that the action with the preferred outcome becomes more likely under $\hat{u}_c, u_o$. In particular she can do so without having to use her (likely imperfect) prior model over possible companies $p(U_o)$. She can use the counterfactual predictions discussed above instead to learn from her experience. We use this insight for counterfactual policy evaluation and search below.

# 3 OFF-POLICY EVALUATION: MODEL-FREE, MODEL-BASED AND COUNTERFACTUAL

To explain how counterfactual reasoning in SCMs can be used for policy search, we first consider the simpler problem of policy evaluation (PE) on off-policy data. The goal of off-policy PE is to determine the value of a policy $\pi$, i.e. its expected return $\mathbb{E}_{\mathfrak{p}^\pi}[G]$, without running the policy itself. We assume that we have data $D = \{\hat{h}_T^i\}_{i=1,\dots,N}$ consisting of logged episodes $\hat{h}_T^i = (\hat{o}_1^i, \hat{a}_1^i, \dots \hat{a}_{T-1}^i, \hat{o}_T^i)$ from running a behavior policy $\mu$. A standard, model-free approach to PE is to use Importance sampling (IS): We can estimate the policy's value as $\sum_i w^i \hat{G}^i$, where $\hat{G}^i$ is the empirical return of $\hat{h}_T^i$ and $w^i \propto \frac{\mathfrak{p}^\pi(\hat{h}_T^i)}{\mathfrak{p}^\mu(\hat{h}_T^i)}$ are importance weights. However, if the trajectory densities $\mathfrak{p}^\pi$ and $\mathfrak{p}^\mu$ are very different, then this estimator has large variance. In the extreme case, IS can be useless if the support of $\mathfrak{p}^\mu$ does not contain that of $\mathfrak{p}^\pi$, irrespective of how much data from $\mathfrak{p}^\mu$ is available.

If we have access to a model $\mathcal{M}$, then we can evaluate the policy on synthetic data, i.e. we can estimate $\mathbb{E}_{p^\pi}[G]$. This is called model-based policy evaluation (MB-PE). However, any bias in $\mathcal{M}$ propagates from $p^\pi$ to the estimate $\mathbb{E}_{p^\pi}[G]$. In the following, we assume that $\mathcal{M}$ is a SCM and we

---

**Algorithm 1** Counterfactual policy evaluation and search

---
    // Counterfactual inference (CFI)
1: **procedure** CFI(data $\hat{x}_o$, SCM $\mathcal{M}$, intervention $I$, query $X_q$)
2:     $\hat{u} \sim p(u|\hat{x}_o)$      ▷ Sample noise variables from posterior
3:     $p(u) \leftarrow \delta(u - \hat{u})$      ▷ Replace noise distribution in $p$ with $\hat{u}$
4:     $f_i \leftarrow f_i^I$      ▷ Perform intervention $I$
5:     **return** $x_q \sim p^{\mathrm{do}(I)}(x_q|\hat{u})$      ▷ Simulate from the resulting model $\mathcal{M}_{\hat{x}_o}^I$
6: **end procedure**

    // Counterfactual Policy Evaluation (CF-PE)
7: **procedure** CF-PE(SCM $\mathcal{M}$, policy $\pi$, replay buffer $D$, number of samples $N$)
8:     **for** $i \in \{1, \dots N\}$ **do**
9:       $\hat{h}_T^i \sim D$      ▷ Sample from the replay buffer
10:       $g_i = \mathrm{CFI}(\hat{h}_T^i, \mathcal{M}, I(\mu \to \pi), G)$      ▷ Counterfactual evaluation of return
11:     **end for**
12:     **return** $\frac{1}{N} \sum_{i=1}^N g_i$
13: **end procedure**

    // Counterfactually-Guided Policy Search (CF-GPS)
14: **procedure** CF-GPS(SCM $\mathcal{M}$, initial policy $\pi^0$, number of trajectory samples $N$)
15:     **for** $k = 1, \dots$ **do**
16:       **if** sometimes **then**
17:         $\mu \leftarrow \pi^k$      ▷ Update behavior policy
18:       **end if**
19:       **for** $i = 1, \dots, N$ **do**
20:         $\hat{h}_T^i \sim \mathfrak{p}^\mu$      ▷ Get off-policy data from the true environment
21:         $\tau^i = \mathrm{CFI}(\hat{h}_T^i, \mathcal{M}, I(\mu \to \pi^\lambda), \mathcal{T})$      ▷ Counterfactual rollouts under planner
22:       **end for**
23:     $\pi^k \leftarrow$ policy improvement on trajectories $\tau^{i=1,\dots,N}$ using eqn. 1
24:     **end for**
25: **end procedure**

---

show that we can use counterfactual reasoning for off-policy evaluation (CF-PE). As the main result for this section, we argue that we expect CF-PE to be less biased than MB-PE, and we illustrate this point with experiments.

## 3.1 COUNTERFACTUAL OFF-POLICY EVALUATION

Naive MB-PE with a SCM $\mathcal{M}$ simply consist of sampling the scenarios $U \sim P_U$ from the prior, and then simulating a trajectory $\tau$ from the functions $f_i$ and computing its return. However, given data $D$ from $\mathfrak{p}^\mu$, our discussion of counterfactual inference in SCMs suggests the following alternative strategy: Assuming no model mismatch, i.e. $\mathfrak{p}^\mu = p^\mu$, we can regard the task of off-policy evaluation of $\pi$ as a counterfactual query with data $\hat{h}_T^i$, intervention $I(\mu \to \pi)$ and query variable $G$. In other words, instead of sampling from the prior as in MB-PE, we are free to the scenarios from the posterior $u^i \sim p^\mu(\cdot|\hat{h}_T^i)$. The algorithm is given in Algorithm 1. Lemma 1 guarantees that this results in an unbiased estimate:

**Corollary 2** (CF-PE is unbiased). *Assuming no model mismatch, CF-PE is unbiased.*

Furthermore, Corollary 1 allows us to also sample some of the noise variables from the prior instead of the posterior, we can e.g. randomize the counterfactual actions by re-sampling the action noise $U_a$.

**Motivation** When should one prefer CF-PE over the more straightforward MB-PE? Assuming a perfect model, Corollary 2 states that both yield the same answer in expectation for perfect models. For imperfect models however, these algorithms can differ substantially. MB-PE relies on purely synthetic data, sampled from the noise distribution $p(U)$. In practice, this is usually approximated by a parametric density model, which can lead to under-fitting in case of complex distributions. This is a well-known effect in generative models with latent variables: In spite of recent research progress, e.g. models of natural images are still unable to accurately model the variability of the true data

(Gregor et al., 2016). In contrast, CF-PE samples from the posterior $N^{-1} \sum_{i=1}^{N} p^{\mu}(U | \hat{h}_T^i)$, which has access to strictly more information than the prior $p(U)$ by taking into account additional data $\hat{h}_T^i$. This semi-nonparametric distribution can help to de-bias the model by effectively winnowing out parts of the domain of $U$ which do not correspond to any real data. We substantiate this intuition with experiments below; a concrete illustration for the difference between the prior and posterior / counterfactual distribution is given in fig. 4 in the appendix and discussed in appendix D. Therefore, we conclude that we expect CF-PE to outperform MB-PE, if the transition and reward kernels $f_{st}$ are accurate models of the environment dynamics, but if the marginal distribution over the noise sources $P_U$ is difficult to model.

## 3.2 EXPERIMENTS

**Environment**   As an example, we use a partially-observed variant of the SOKOBAN environment, which we call PO-SOKOBAN. The original SOKOBAN puzzle environment was described in detail by Racanière et al. (2017); we give a brief summary here. The agent is situated in a $10 \times 10$ grid world and its five actions are to move to one of four adjacent tiles and a NOOP. In our variant, the goal is to push all three boxes onto the three targets. As boxes cannot be pulled, many actions result irreversibly in unsolvable states. Episodes are of length $T = 50$, and pushing a box onto a target yields a reward of 1, removing a box from a target yields $-1$, and solving a level results in an additional reward of 10. The state of the environment consists in a $10 \times 10$ matrix of categorical variables taking values in $\{0, \dots, 6\}$ indicating if the corresponding tile is empty, a wall, box, target, agent, or a valid combinations thereof (box+target and agent+target). In order to introduce partial observability, we define the observations as the state corrupted by i.i.d. (for each tile and time step) flipping each categorical variable to the "empty" state with probability 0.9. Therefore, the state of the game is largely unobserved at any given time, and a successful agent has to integrate observations over tens of time steps. Initial states $U_{s1}$, also called *levels*, which are the scenarios in this environment, are generated randomly by a generator algorithm which guarantees solvability (i.e. all boxes can be pushed onto targets). The environment is visualized in fig. 3 in the appendix.

Given the full state of PO-SOKOBAN, the transition kernel is deterministic and quite simple as only the agent and potentially an adjacent box moves. Inferring the belief state, i.e. the distribution over states given the history of observations and actions, can however range from trivial to very challenging, depending on the amount of available history. In the limit of a long observed history, every tile is eventually observed and the belief state concentrates on a single state (the true state) that can be easily inferred. With limited observed history however, inferring the posterior distribution over states (belief state) is very complex. Consider e.g. the situation in the beginning of an episode (before pushing the first box). Only the first observation is available, however we know that all PO-SOKOBAN levels are initially guaranteed to be solvable and therefore satisfy many combinatorial constraints reflecting that the agent is still able to push all boxes onto targets. Learning a compact parametric model of the initial state distribution from empirical data is therefore difficult and likely results in large mismatch between the learned model and the true environment.

**Results**   To illustrate the potential advantages of CF-PE over MB-PE we perform policy evaluation in the PO-SOKOBAN environment. We first generate a policy $\pi$ that we wish to evaluate, by training it using a previously-proposed distributed RL algorithm (Espeholt et al., 2018). The policy is parameterized as a deep, recurrent neural network consisting of a 3-layer deep convolutional LSTM (Xingjian et al., 2015) with 32 channels per layer and kernel size of 3. To further increase computational power, the LSTM ticks twice for each environment step. The output of the agent is a value function and a softmax yielding the probabilities of taking the 5 actions. In order to obtain an SCM of the environment, for the sake of simplicity, we assume that the ground-truth transition, observation and reward kernels are given. Therefore the only part of the model that we need to learn is the distribution $p(U_{s1})$ of initial states $S_1 = U_{s1}$ (for regular MB-PE), and the density $p(U_{s1} | \hat{h}_t^i)$ for inferring levels in hindsight for CF-PE. We vary the amount of true data $t$ that we condition this inference on, ranging from $t = 0$ (no real data, equivalent to MB-PE) to $t = T = 50$ (a full episode of real data is used to infer the initial state $U_{s1}$). We train a separate model for each $t \in \{0, 5, 10, 20, 30, 40, 50\}$. To simplify model learning, both models were given access to the unobserved state during training, but not at test time. The models are chosen to be powerful, multi-layer, generative DRAW models (Gregor et al., 2015) trained by approximate maximum likelihood learning (Kingma & Welling, 2013; Rezende et al., 2014). The models take as input the (potentially

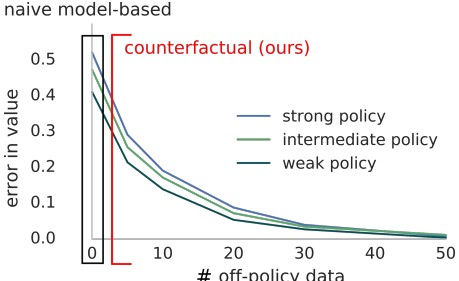 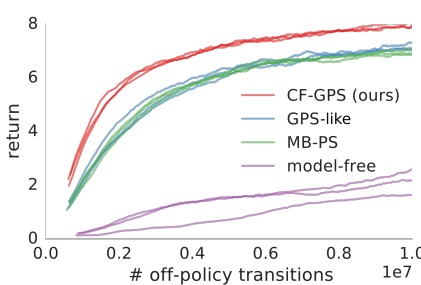

Figure 2: Experimental results on PO-SOKOBAN environment. **Left: Policy evaluation.** Policy evaluation error decreases with amount of off-policy data available (in #transitions per episode) for inferring scenarios (levels) $U_{s1}$ that are used for counterfactual evaluation. No data (data points on the very left) corresponds to standard model-based policy evaluation (MB-PE), yielding large errors, whereas Counterfactual policy evaluation yields more accurate results. This holds for all three policies with different true performances. **Right: Policy search.** Counterfactually-Guided Policy Search (CF-GPS) outperforms a naive model-based RL (MB-PS) algorithm as well as a version of standard Guided Policy Search ('GPS-like') on PO-SOKOBAN.

empty) data $\hat{h}_t^i$ summarized by a backward RNN (a standard convolutional LSTM model with 32 units). The model is shown in fig. 3 in the appendix and additional details are given in appendix C. The data $\hat{h}_T^i$ was collected under a uniform random policy $\mu$. For all policy evaluations, we use $\approx > 10^5$ levels $u^i$ from the inferred model. In order to evaluate policies of different proficiency, we derive from the original (trained) $\pi$ three policies $\pi_0, \pi_1, \pi_2$ ranging from almost perfect to almost random performance by introducing additional stochasticity during action selection.

The policy evaluation results are shown in fig. 2. We found that for $t = 0$, in spite of extensive hyper-parameter search, the model $p(U_{s1})$ was unable to accurately capture the marginal distribution of initial levels in PO-SOKOBAN. As argued above, a solvable level satisfies a large number of complex constraints that span the entire grid world, which are hard for a parametric model to capture. Empirically, we found that the model mismatch manifested itself in samples from $p(U_{s1})$ not being well-formed, e.g. not solvable, and hence the performance of the policies $\pi_i$ are very different on these synthetic levels compared to levels sampled form $\mathfrak{p}$. However, inferring levels from full observed episodes i.e. $p(U_{s1}|\hat{h}_{50}^i)$ was reliable, and running $\pi$ on these resulted in accurate policy evaluation. The figure also shows the trade-off between policy evaluation accuracy and the amount of off-policy data for intermediate amounts of the data $\hat{h}_t^i$. We also want to emphasize that in this setting, model-free policy evaluation by IS fails. The uniform behavior policy $\mu$ was too different from $\pi^i$, resulting in a relative error $> 0.8$ for all $i = 1, 2, 3$.

## 4 OFF-POLICY IMPROVEMENT: COUNTERFACTUALLY-GUIDED POLICY SEARCH

In the following we show how we can leverage the insights from counterfactual policy evaluation for policy search. We commence by considering a model-based RL algorithm and discuss how we can generalize it into a counterfactual algorithm to increase its robustness to model mismatch. We chose a particular algorithm to start from to make a connection to the previously proposed Guided Policy Search algorithm (Levine & Koltun, 2013; Levine & Abbeel, 2014), but we think a larger class of MBRL algorithms can be generalized in an analogous manner.

### 4.1 STARTING POINT: VANILLA MODEL-BASED RL WITH RETURN WEIGHTED REGRESSION

We start from the following algorithm. We assume we have a model $\mathcal{M}$ of the environment with trajectory distribution $p^\pi$. Our current policy estimate $\pi^k$ is improved at iteration $k$ using return-weighted regression:

$$\pi^{k+1} = \arg\max_\pi \int \exp(G(\tau)) p^{\pi^k}(\tau) \log p^\pi(\tau) \, d\tau,$$

where $G(\tau)$ is the return of trajectory $\tau$. This policy improvement step can be motivated by the framework of RL as variational inference (Toussaint, 2009) and is equivalent to minimizing the KL divergence to a trajectory distribution $\propto \exp(G)p^{\pi^k}$ which puts additional mass on high-return trajectories. Although not strictly necessary for our exposition, we also allow for a dedicated proposal distribution over trajectories $p^\lambda(\tau)$, under a policy $\lambda$. We refer to $\lambda$ as a *planner* to highlight that it could consist of a procedure that solves episodes starting from arbitrary, full states $s_1$ sampled form the model, by repeatedly calling the model transition kernel, e.g. a search procedure such as MCTS (Browne et al., 2012) or an expert policy. Concretely, we optimize the following finite sample objective:

$$\pi^{k+1} \quad = \quad \arg\max_\pi \sum_{i=1}^N \exp(G^i(\tau^i)) \frac{p^{\pi^k}(\tau^i)}{p^\lambda(\tau^i)} \log p^\pi(\tau^i), \qquad \tau^i \sim p^\lambda. \tag{1}$$

We refer to this algorithm as model-based policy search (MB-PS). It is based on model rollouts $\tau^i$ spanning entire episodes. An alternative would be to consider model rollouts starting from states visited in the real environment (if available). Both versions can be augmented by counterfactual methods, but for the sake of simplicity we focus on the simpler MB-PS version detailed above (also we did not find significant performance differences experimentally between both versions).

## 4.2 INCORPORATING OFF-POLICY DATA: COUNTERFACTUALLY-GUIDED POLICY SEARCH

Now, we assume that the model $\mathcal{M}$ is an SCM. Based on our discussion of counterfactual policy evaluation, it is straightforward to generalize the MB-PS described above by anchoring the rollouts $\tau^i$ under the model $p^\lambda$ in off-policy data $D$: Instead of sampling $\tau^i$ directly from the prior $p^\lambda$, we draw them from counterfactual distribution $p^{\lambda|\hat{h}_T^i}$ with data $\hat{h}_T^i \sim D$ from the replay buffer, i.e. instead of sampling the scenarios $U$ from the prior we infer them from the given data. Again invoking Lemma 1, this procedure is unbiased under no model mismatch. We term the resulting algorithm Counterfactually-Guided Policy Search (CF-GPS), and it is summarized in Algorithm 1. The motivation for using CF-GPS over MB-PS is analogous to the advantage of CF-PE over MB-PE discussed in sec. 3.1. The policy $\pi$ in CF-GPS is optimized on rollouts $\tau^i$ that are grounded in data $\hat{h}_T^i$ by sampling them from the counterfactual distribution $p^{\lambda|\hat{h}_T^i}$ instead of the prior $p^\lambda$. If this prior is difficult to model, we expect the counterfactual distribution to be more concentrated in regions where there is actual mass under the true environment $\mathfrak{p}^\lambda$.

## 4.3 EXPERIMENTS

We evaluate CF-GPS on the PO-SOKOBAN environment, using a modified distributed actor-learner architecture based on Espeholt et al. (2018): Multiple actors (here 64) collect real data $\hat{h}_T$ by running the behavior policy $\mu$ in the true environment $\mathfrak{p}$. As in many distributed RL settings, $\mu$ is chosen to be a copy of the policy $\pi$, often slightly outdated, so the data must be considered to be off-policy. The distribution $p(U_{s1}|\hat{h}_T)$ over levels $U_{s1}$ is inferred from the data $\hat{h}_T$ using from the model $\mathcal{M}$. We sample a scenario $U_{s1}$ for each logged episode, and simulate 10 counterfactual trajectories $\tau^{1,\dots,10}$ under the planner $\lambda$ for each such scenario. Here, for the sake of simplicity, instead of using search, the planner was assumed to be a mixture between $\pi$ and a pre-trained expert policy $\lambda_e$, i.e. $\lambda = \beta\lambda_e + (1-\beta)\pi$. The schedule $\beta$ was set to an exponentially decaying parameter with time constant $10^5$ episodes. The learner performs policy improvement on $\pi$ using $\tau^{1,\dots,10}$ according to eqn. 1. $\mathcal{M}$ was trained online, in the same way as in sec. 3.2. $\lambda$ and $\pi$ were parameterized by deep, recurrent neural networks with the same architecture described in sec. 3.2.

We compare CF-GPS with the vanilla MB-PS baseline described in sec. 4.1 (based on the same number of policy updates). MB-PS differs from CF-GPS by just having access to an unconditional model $p(U_{s1}|\emptyset)$ over initial states. We also consider a method which conditions the scenario model $p(U_{s1}|o_1)$ on the very first observation $o_1$, which is available when taking the first action and therefore does not involve hindsight reasoning. This is more informed compared to MB-PS; however due to the noise on the observations, the state is still mostly unobserved rendering it very challenging to learn a good parametric model of the belief state $p(U_{s1}|o_1)$. We refer to this algorithm as Guided Policy Search-like (GPS-like), as it roughly corresponds to the algorithm presented by Levine & Abbeel (2014), as discussed in greater detail in sec. 5. Fig. 2 shows that CF-GPS outperforms these

two baselines. As expected from the policy evaluation experiments, initial states sampled from the models for GPS and MB-PS are often not solvable, yielding inferior training data for the policy $\pi$. In CF-GPS, the levels are inferred from hindsight inference $p(U_1|\hat{h}_T)$, yielding high quality training data. For reference, we also show a policy trained by the model-free method of Espeholt et al. (2018) using the same amount of environment data. Not surprisingly, CF-GPS is able to make better use of the data compared to the model-free baseline as it has access to the true transition and reward kernels (which were not given to the model-free method).

## 5 RELATED WORK

Bottou et al. (2013) provide an in-depth discussion of applying models to off-policy evaluation. However, their and related approaches (Li et al., 2015; Jiang & Li, 2015; Swaminathan & Joachims, 2015; Nedelec et al., 2017; Atan et al., 2016; Thomas & Brunskill, 2016) such as doubly-robust estimators, rely on Importance Sampling (IS), also called Propensity Score method. Although some of these algorithms are also termed counterfactual policy evaluation, they are not counterfactual in the sense used in this paper, where noise variables are inferred from logged data and reused to evaluate counterfactual actions. Hence, they are dogged by high variance in the estimators common to IS, although recent work aims to address this (Munos et al., 2016; Guo et al., 2017). Model-based methods for off-policy evaluation have recently been improved to account for the distribution shift between the data-collecting policy and the policy to be evaluated (Johansson et al., 2016; Liu et al., 2018). Recently (Andrychowicz et al., 2017) proposed the Hindsight Experience Replay (HER) algorithm for learning a family of goal directed policies. In HER one observes an outcome in the true environment, which is kept fixed, and searches for the goal-directed policy that should have achieved this goal in order to positively reinforce it. Therefore, this algorithm is complementary to CF-GPS where we search over alternative outcomes for a given policy. Our CF-GPS algorithm is inspired by and extends work presented by Abbeel et al. (2006) on a method for de-biasing weak models by estimating additive terms in the transition kernel to better match individual, real trajectories. The resulting model, which is a counterfactual distribution in the terminology used in our paper, is then used for model-based policy improvement. Our work generalizes this approach and highlights conceptual connections to causal reasoning. Furthermore, we discuss the connection of CF-GPS to two classes of RL algorithms in greater detail below.

**Guided Policy Search (GPS)** CF-GPS is closely related to GPS, in particular we focus on GPS as presented by Levine & Abbeel (2014). Consider CF-GPS in the fully-observed MDP setting where $O_t = S_t$. Furthermore, assume that the SCM $\mathcal{M}$ is structured as follows: Let $S_{t+1} = f_s(S_t, A_t, U_{st})$ be a linear function in $(S_t, A_t)$ with coefficients given by $U_{st}$. Further, assume an i.i.d. Gaussian mixture model on $U_{st}$ for all $t$. As the states are fully observed, the inference step in the CFI procedure simplifies: we can infer the noise sources $\hat{u}_{st}$ (samples or MAP estimates), i.e. the unknown linear dynamics, from pairs of observed, true states $\hat{s}_t, \hat{s}_{t+1}$. Furthermore assume that the reward is a quadratic function of the state. Then, the counterfactual distribution $p^\lambda(\tau|\hat{u})$ is a linear quadratic regulator (LQR) with time-varying coefficients $\hat{u}$. An appropriate choice for the planner $\lambda$ is the optimal linear feedback policy for the given LQR, which can be computed exactly by dynamic programming.

**Observation 1.** *In the MDP setting, CF-GPS with a linear SCM and a dynamic programming planner for LQRs $\lambda$ is equivalent to GPS.*

Another perspective is that GPS is the counterfactual version of the MB-PS procedure from sec. 4.1:

**Observation 2.** *In the MDP setting with a linear SCM and a dynamic programming planner for LQRs $\lambda$, GPS is the counterfactual variant of the MB-PS procedure outlined above.*

The fact that GPS is a successful algorithm in practice shows that the 'grounding' of model-based search / rollouts in real, off-policy data afforded by counterfactual reasoning massively improves the naive, 'prior sample'-based MB-PS algorithm. These considerations also suggest when we expect CF-GPS to be superior compared to regular GPS: If the uncertainty in the environment transition $U_{st}$ cannot be reliably identified from subsequent pairs of observations $\hat{o}_t, \hat{o}_{t+1}$ alone, we expect benefits of inferring $U_{st}$ from a larger context of observations, in the extreme case from the entire history $\hat{h}_T$ as described above.

**Stochastic Value Gradient methods** There are multiple interesting connections of CF-GPS to Stochastic Value Gradient (SVG) methods (Heess et al., 2015). In SVG, a policy $\pi$ for a MDP is learned by gradient ascent on the expected return under a model $p$. Instead of using the score-function estimator, SVG relies on a reparameterization of the stochastic model and policy (Kingma & Welling, 2013; Rezende et al., 2014). We note that this reparameterization casts $p$ into an SCM. As in GPS, the noise sources $U_{st}$ are inferred from two subsequent observed states $\hat{s}_t, \hat{s}_{t+1}$ from the true environment, and the action noise $U_{at}$ is kept frozen. As pointed out in the GPS discussion, this procedure corresponds to the inference step in a counterfactual query. Given inferred values $u$ for $U$, gradients $\partial_\theta G$ of the return under the model are taken with respect to the policy parameters $\theta$. We can loosely interpret these gradients as $2 \dim(\theta)$ counterfactual policy evaluations of policies $\pi(\theta \pm \Delta\theta_i)$ where a single dimension $i$ of the parameter vector $\theta$ is perturbed.

## 6 DISCUSSION

Simulating plausible synthetic experience de novo is a hard problem for many environments, often resulting in biases for model-based RL algorithms. The main takeaway from this work is that we can improve policy learning by evaluating counterfactual actions in concrete, past scenarios. Compared to only considering synthetic scenarios, this procedure mitigates model bias. However, it relies on some crucial assumptions that we want to briefly discuss here. The first assumption is that off-policy experience is available at all. In cases where this is e.g. too costly to acquire, we cannot use any of the proposed methods and have to exclusively rely on the simulator / model. We also assumed that there are no additional hidden confounders in the environment and that the main challenge in modelling the environment is capturing the distribution of the noise sources $p(U)$, whereas we assumed that the transition and reward kernels given the noise is easy to model. This seems a reasonable assumption in some environments, such as the partially observed grid-world considered here, but not all. Probably the most restrictive assumption is that we require the inference over the noise $U$ given data $\hat{h}_T$ to be sufficiently accurate. We showed in our example, that we could learn a parametric model of this distribution from privileged information, i.e. from joint samples $u, h_T$ from the true environment. However, imperfect inference over the scenario $U$ could result e.g. in wrongly attributing a negative outcome to the agent's actions, instead environment factors. This could in turn result in too optimistic predictions for counterfactual actions. Future research is needed to investigate if learning a sufficiently strong SCM is possible without privileged information for interesting RL domains. If, however, we can trust the transition and reward kernels of the model, we can substantially improve model-based RL methods by counterfactual reasoning on off-policy data, as demonstrated in our experiments and by the success of Guided Policy Search and Stochastic Value Gradient methods.

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

# A  PROOFS

## A.1  PROOF OF LEMMA 1

*Proof.* We start from the fact that the density over noise sources $U$ remains the same for every intervention $I$ as $U$ are root nodes in $\mathcal{G}$:

$$p^{\mathrm{do}(I)}(u) = p(u).$$

This leads to:

$$
\begin{aligned}
p^{\mathrm{do}(I)}(x) &= \int p^{\mathrm{do}(I)}(x|u)p^{\mathrm{do}(I)}(u)\,du \\
&= \int p^{\mathrm{do}(I)}(x|u)p(u)\,du \\
&= \int p^{\mathrm{do}(I)}(x|u)\left(\int p(\hat{x}_o, u)d\hat{x}_o\right)\,du \\
&= \int\int p^{\mathrm{do}(I)}(x|u)\,p(u|\hat{x}_o)p(\hat{x}_o)\,du\,d\hat{x}_o \\
&= \mathbb{E}_{\hat{x}_o \sim p}\Big[\int p^{\mathrm{do}(I)}(x|u)p(u|\hat{x}_o)\,du\Big] \\
&= \mathbb{E}_{\hat{x}_o \sim p}[p^{\mathrm{do}(I)|\hat{x}_0}(x)].
\end{aligned}
$$

$\square$

## A.2  PROOF OF COROLLARY 1

*Proof.* Given two sets $\mathrm{CF} \subset \{1, \ldots, N\}$ and $\mathrm{Prior} \subset \{1, \ldots, N\}$ with $\mathrm{CF} \cap \mathrm{Prior} = \emptyset$ and $\mathrm{CF} \cup \mathrm{Prior} = \{1, \ldots, N\}$, we define $u_{\mathrm{CF}} = (u_n)_{n \in \mathrm{CF}}$ and $u_{\mathrm{Prior}} = (u_n)_{n \in \mathrm{Prior}}$. By construction, the scenarios $U$ are independent under the prior, i.e. $p(u) = \prod_{n=1}^N p(u_n)$. Therefore $u_{\mathrm{CF}}$ and $u_{\mathrm{Prior}}$ are independent. We can write:

$$
\begin{aligned}
p(u) &= \left(\prod_{n \in \mathrm{CF}} p(u_n)\right)\left(\prod_{n \in \mathrm{Prior}} p(u_n)\right) \\
&= p(u_{\mathrm{CF}})p(u_{\mathrm{Prior}}).
\end{aligned}
$$

Following the arguments from Lemma 1, the averaged inference distribution is equal to the prior $p(u) = \mathbb{E}_{\hat{x}_o \sim p}[p(u|\hat{x}_o)]$. This also holds for any subset of the variables $u$, in particular for for $p(u_{\mathrm{CF}})$. Hence:

$$
\begin{aligned}
p(u) &= p(u_{\mathrm{CF}})p(u_{\mathrm{Prior}}) \\
&= \mathbb{E}_{\hat{x}_o \sim p}[p(u_{\mathrm{CF}}|\hat{x}_o)]p(u_{\mathrm{Prior}}).
\end{aligned}
$$

$\square$

# B  DETAILS ON CASTING A POMDP INTO SCM FORM

**Lemma 2** (Auto-regressive uniformization aka Reparametrization)**.** *Consider random variables $X_1, \ldots, X_N$ with joint distribution $P$. There exist functions $f_n$ for $n \in \{1, \ldots, N\}$ such that with independent random variables $U_n \sim \mathrm{Uniform}[0, 1]$ the random variables $X'$ equal $X$ in distribution, i.e. $X' \stackrel{d}{=} X$, where $X' = (X'_1, \ldots, X'_n)$ are defined as:*

$$X'_n := f_n(U_n, X'_{<n}).$$

*Proof.* We construct the functions $f_n$ by induction on $n$. Consider the conditional distribution $P_{X_n|X_{<n}}$. For fixed $X_{<n}$, denote its CDF with $F_{n,X_{<n}}$. We construct a random variable $X' := F_{n,X_{<n}}^{-1}(U_n)$ with $U_n \sim \mathrm{Uniform}[0, 1]$ independent from $X_{<n}$ and $U_{<n}$. By virtue of the inverse-CDF method, we have $X'|X_{<n} \stackrel{d}{=} X|X_{<n}$. Therefore, $f_n(U_n, X_{<n}) := F_{n,X_{<n}}^{-1}(U_n)$ satisfies the above lemma. $\square$

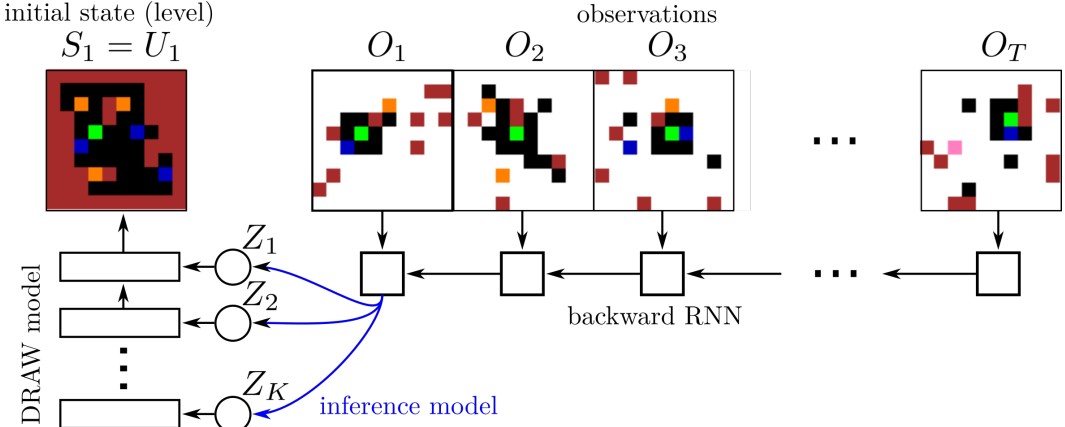

Figure 3: **Top: PO-SOKOBAN**. Shown on the left is a procedurally generated initial state. The agent is shown in green, boxes in yellow, targets in blue and walls in red. The agent does not observe this state but a sequence of observations, which are masked by iid noise with 0.9 probability, except a 3x3 window around the agent. **Bottom: Inference model.** For counterfactual inference in PO-SOKOBAN, we need the (approximate) inference distribution $p(U_{s1}|\hat{h}_T)$ over the initial state $U_{s1} = S_1$, conditioned on the history of observations $\hat{h}_T$. We model this distribution using a DRAW generative model with latent variables $Z$, which are conditioned on the output of a backward RNN summarizing the observation history.

## C    MODEL ARCHITECTURE

We assume that we are given the true transition and reward kernels. As the transitions are deterministic in PO-SOKOBAN, the only part of the model that remains to be identified is the initial state distribution $p(U_{s1})$. We learned this model from data using a the DRAW model (Gregor et al., 2015), which is a parametric, multi-layer, latent variable, neural network model for distributions. For our purposes we chose the convolutional DRAW architecture proposed by (Gregor et al., 2016). First, the observation data is summarized by a convolutional LSTM with 32 hidden units and kernel size of 3. The resulting final LSTM state is fed into a conditional Gaussian prior over the latent variables $Z_{k=1,...,8}$ of the 8-layer conv-DRAW model. Each layer has 32 hidden layers and the canvas had 7 layers, corresponding to the 7 channels of the categorical $U_{s1} \in \{0,1\}^{10 \times 10 \times 7}$ that we wish to model. The model (together with the backward RNN) was trained with the ADAM optimizer (Kingma & Ba, 2014) on the ELBO loss using the reparametrization trick (Kingma & Welling, 2013; Rezende et al., 2014). The mini-batch size was set to 4 and the learning rate to $3e-4$. We want to emphasize that the DRAW latent variables $Z$ are not directly the noise variables $U$ of the SCM, but integrating out these variables yields this distribution $p(U_{s1}|\hat{h}_T) = \int p(U_{s1}|z, \hat{h}_T)p(z|\hat{h}_T)d\hat{h}_T$.

## D    MODEL MISMATCH ANALYSIS

Here we provide some analysis of the DRAW model over the initial state $U_{s1}$, which is the learned part of the SCM $\mathcal{M}$ used for the policy evaluation experiments presented in 3.2. As detailed above, we trained a separate model $p(U_{s1}|\hat{h}_t)$ for each $t = 1, \ldots, 50$ parameterizing the cardinality of the data the model is conditioned on. We analyze three particular models for $t = 0, 1$ and $50$ which we term the unconditional / filtering / smoothing model, as they are conditioned on no data / on data that is available at test time / all data that is available in hindsight. Directly visualizing the distributions $p(U_{s1}|\hat{h}_t)$ for an analysis is difficult as the domain $\{0, \ldots, 6\}^{10 \times 10}$ is high-dimensional and discrete. Instead we focus on the latent variables $Z$ which are learned by DRAW to represent this distribution; by construction, these are jointly Normal, facilitating the analysis. In particular, we compare $p(Z|\hat{h}_t)$ with the inference distribution $q(Z|\hat{u}_{s1})$ conditioned on the true state $\hat{U}_{s1}$. We loosely interpret $q(Z|\hat{u}_{s1})$ as the "true" embedding of the datum $\hat{u}_{s1}$, whereas $p(Z|\hat{h}_t)$ is the learned

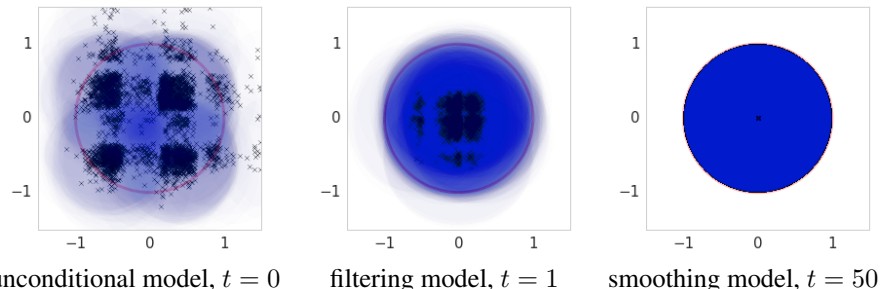

unconditional model, $t = 0$      filtering model, $t = 1$      smoothing model, $t = 50$

Figure 4: Analysis of the model mismatch of the learned inference distributions $p(U_{s1}|\hat{h}_t)$ over the initial PO-SOKOBAN state $U_{s1}$, for three different amounts of observations $t = 0, 1$ and $50$. Shown are two dimensions of the learned latent representation $Z$ of $U_{s1}$. The (whitened) learned prior $p(Z|\hat{h}_t)$ is indicated by a red contour of one standard deviation. The inferred mean embedding of the true levels are show as crosses, and their aggregated density is shown in blue. With increasing amount data that the model is conditioned on, the learned distributions match the data better.

embedding. In a perfect model the prior matches the inference distribution on average:

$$\mathbb{E}_{\hat{h}_t \sim \mathfrak{p}^\mu}[p(Z|\hat{h}_t)] \quad \overset{!}{=} \quad \mathbb{E}_{\hat{u}_{s1} \sim \mathfrak{p}^\mu}[q(Z|\hat{u}_{s1})],$$

i.e. every sample from the prior $p(Z|\hat{h}_t)$ corresponds to real data and vice versa. We visualize the averaged prior $\mathbb{E}_{\hat{h}_t \sim \mathfrak{p}^\mu}[p(Z|\hat{h}_t)]$ and the averaged posterior $\mathbb{E}_{\hat{u}_{s1} \sim \mathfrak{p}^\mu}[q(Z|\hat{u}_{s1})]$ in fig. 4. We show the two dimensions of $Z$ where these distributions have the largest KL divergence. Also, the plots were whitened w.r.t. the averaged prior, i.e. the latter is a spherical Gaussian in the plots, represented by an iso-probability contour corresponding to one standard deviation. The inference distribution for each datum $\hat{u}_{s1}$ is visualized by its mean (cross) and a level set corresponding the one standard deviation or less. For the unconditional model $t = 0$, we find that the distributions are not matched well. In particular, there is a lot of prior mass that sits in regions where there is no or little true data. In the RL setting this results in synthetic data from the model that is unrealistic and training a policy on this data leads to reduced test performance. Also, as apparent from the figure, there is structure in the embedding of the true data, that is not captured at all by the prior. This effect is markedly reduced in the filtering posterior, indicating that the conditional distribution $p(Z|\hat{h}_1)$ already captures the data distribution better. The smoothing model is a very good match to the data. With high probability, all tiles of the game are observed in $\hat{h}_{50}$, enabling the model to perfectly learn the belief state, which collapses in this setting to a single state.