# OpenReview forum: "Woulda, Coulda, Shoulda: Counterfactually-Guided Policy Search"
_ICLR.cc/2019/Conference_

### Official Review · AnonReviewer2 · 2018-11-03
**Interesting approach to relevant problem; nice integration of causal reasoning with RL; experiment setup avoids dealing with some practical challenges**

**Rating:** 7
**Confidence:** 3

**Review:**

Summary:
Proposes Counterfactual Guided Policy Search (CF-GPS), which uses counterfactual inference from sampled trajectories to improve an approximate simulator that is used for policy evaluation. Counterfactual inference is formalized with structural causal models of the POMDP. The method is evaluated in partially-observed Sokoban problems. The dynamics model is assumed known, and a learned model maps observation histories to a conditional distribution on the starting state. CF-GPS outperforms model-based policy search and a "GPS-like" algorithm in these domains. GPS in MDPs is shown to be a particular case of CF-GPS, and a connection is also suggested between stochastic value gradient and CF-GPS.

Review:
The work is an interesting approach to a relevant problem. Related literature is covered well, and the paper is well-written in an approachable, conversational style.

The approach is technically sound and generally presented clearly, with a few missing details. It is mainly a combination of existing tools, but the combination seems to be novel.

The experiments show that the method is effective for these Sokoban problems. A weakness is that the setting is very "clean" in several ways. The dynamics and rewards are assumed known and the problem itself is deterministic, so the only thing being inferred in hindsight is the initial state. This could be done without all of the machinery of CF-GPS. I realize that the CF-GPS approach is domain-agnostic, but it would be useful to see it applied in a more general setting to get an idea of the practical difficulties. The issue of inaccurate dynamics models seems especially relevant, and is not addressed by the Sokoban experiment. It's also notable that the agent cannot affect any of the random outcomes in this problem, which I would think would make counterfactual reasoning more difficult.

Comments / Questions:
* Please expand on what "auto-regressive uniformization" is and how it ensures that every POMDP can be expressed as an SCM
* What is the prior p(U) for the experiments?
* "lotion-scale" -> "location-scale"

Pros:
* An interesting and well-motivated approach to an important problem
* Interesting connections to GPS in MDPs

Cons:
* Experimental domain does not "exercise" the approach fully; the counterfactual inference task is limited in scope and the dynamics and rewards are deterministic and assumed known
* Work may not be easily reproducible due to the large number of pieces and incomplete specification of (hyper-)parameter settings

---

> ### Author Response · Authors · 2018-11-19
> **Re review**
>
> We added a paragraph on the “auto-regressive uniformization” in the appendix, showing how any joint distribution over random variables can be converted into independent noise variables and deterministic functions. Please also see our reply to reviewer 1.
>
> Concerning the choice of prior $p(u)$ in the experiments: $U$ was defined as the initial state or “level” of the environment. The prior, as well as the posterior, were chosen to be DRAW latent variable models. The only difference between these models was that the one encoding the posterior was conditioned on observed data $h$. This parametrization is also discussed in the appendix D.

---

### Official Review · AnonReviewer1 · 2018-11-08
**Interesting ideas; unclear if assumptions are too strong**

**Rating:** 7
**Confidence:** 3

**Review:**

Summary: by assuming a correct, strongly factored environment model, improved estimators useful for policy search can be derived by "counterfactual reasoning", where data sampled from experience is used to refine initial conditions in the model; this translates into improved estimators of policy values, which improves policy search.

Major comments:

I enjoyed this paper.  I think that model-based RL deserves more work, and I think that this is a simple, reasonably workable approach with some nice theoretical benefits.  I like the idea of SCMs; I like the idea of counterfactual reasoning; I like the idea of leveraging models in this unique way.

On the negative side, I felt that the paper makes some rather strong assumptions - specifically, that the agent has access to a perfect model with no mismatch, and that the model decomposes neatly into noise variables plus deterministic functions.  Given such a model, one wonders if there are other techniques, say, from classical planning, that could also be used for some sort of policy search.

I have a few questions about approximations.  First, I see that probabilistic inference is a core element of each algorithm (where p(u|h) must be computed).  For large, complex models, I assume this must be approximate inference.  This leads naturally to questions about accuracy (does approximate inference result in biased estimators? [probably yes]), efficacy (do the inaccuracies inherent in approximate inference outweigh the benefits of using p(u|h) vs. p(u)?) and scalability (how large of a model can we reasonably cope with before degradation is unacceptable, or no better than non-CF algorithms?).  As far as I can tell, none of this was addressed in the paper, although I do not expect every paper to answer every question; this is a first step.

I wish the experiments were a little more varied.  The experimental results really only show marginal improvement in one small task.  While I understand that this is not an empirical paper, neither does it fit strongly into the category of "theory paper".  For example, there are no theory results indicating what sort of benefit we might expect from using the methods outlined here, and in the absence of such theory, we might reasonably look to various experiments to demonstrate its effectiveness.

Pros:
+ Integration with SCMs is interesting
+ Counterfactual variants of algorithms are clearly motivated and interesting
+ Paper is generally well-written

Cons:
- Assumption that the agent is given a model with no mismatch is very strong
- Model class (noise variables + deterministic functions) seems potentially restrictive
- Questions about impact of approximate inference
- Experiments could have been more varied

---

> ### Author Response · Authors · 2018-11-19
> **Re review**
>
> We agree that our algorithm makes strong assumptions about the model, and that we have not yet studied theoretically or experimentally the important question raised by the reviewer of how violations of these assumptions influence performance.
>
> We want clarify however, that the assumption of the model class consisting of deterministic functions and independent noise variables is not restrictive in itself, any joint probability over random variables can be written in this way by iteratively applying the “inverse-CDF” method. For a joint Gaussian for example, this corresponds to sampling one variable at a time (conditioned on the previous ones) by sampling an RV uniformly in [0,1], passing it through the inverse standard-Gaussian CDF and scaling it with the conditional standard deviation and adding the conditional mean. We added a paragraph in the appendix to clarify this point.

---

### Official Review · AnonReviewer4 · 2018-11-12
**Interesting problem and approach; more experimental domains and careful analysis on experiment results would be appreciated.**

**Rating:** 7
**Confidence:** 2

**Review:**

Summary:

This paper proposes a policy evaluation and search method assisted by a counterfactual model, in contrast previous work using vanilla (non-causal) models. With “no model mismatch” assumption the policy evaluation estimator is unbiased. Empirically, the paper compares Guided Policy Search with counterfactual model (CF-GPS) with vanilla GPS, model based RL algorithm and show benefit in terms of (empirical) sample complexity.

Main comments:

This paper studies several interesting problems: 1) policy learning with off-policy data; 2) model based RL and how to use model to help policy learning. By capturing a nice connection between causal models and MDP/POMDP model with off-policy data, this paper can leverage SCMs to help the model guided policy search in POMDP. The combination of those ideas is novel and enjoyable.

On the negative side, I find I met several confused points as a reader with more RL background and less causal inference background. It would be better if the authors could clarify what is the prior distribution P(u) and posterior distribution P(u|h) exactly means in terms of CF-PE algorithm and MB-PE algorithm. I would also appreciate if a more detailed proof of corollary 1 and 2 are included in the appendix, and a higher level intuition/justification about those two results in main body. Maybe I am missing these points due to my limited background in causal inference, but I think those clarification can definitely be helpful for RL audience without that much knowledge in causal inference.

The main theoretical result seems to be based on the assumption of no model mismatch, and I guess here how the model is estimated from sample are ignored, unless I missed anything. Thus I assume the main contribution of this paper should be algorithmic and empirical. I expect to see the empirical study in more domains with more informative results about how this CF model get the benefit of sampling from p(u|h) rather than p(u) (as an evidence to support motivation paragraph on page 5).

---

> ### Author Response · Authors · 2018-11-19
> **Re review**
>
> For improved readability, we added a proof for corollary 1 in the appendix. Corollary 2 is a direct application of lemma 1 to the SCM prepresentation of a POMDP.
>
> Concerning the difference between $p(u)$ vs $(u\vert h)$:
> Standard model based RL (MBRL) algorithms usually try to learn a model over unobserved variables $U$ of the environment. If there is uncertainty over these given the observations, then a natural approach for MBRL would to learn a distribution, ie prior $p(u)$. At model test time, one usually samples from this prior to generate rollouts for policy evaluation (or learning). This corresponds to the MB-PE procedure. We propose, instead of sampling from the prior, given concrete observed data $h$, sample from the posterior $p(u\vert h)$, yielding the CF-PE algorithm. As argued in the paper, $p(u\vert h)$ should be easier to learn than $p(u)$. We hope the “motivation” paragraphs in the introduction and Ch2 can give an intuitive understanding of the difference.

---

### Author Response · Authors · 2018-11-19
**Reply to reviewers**

We thanks the reviewers for the their thoughtful comments, some of which we address individually below.

Generally, we want to emphasize that the main contribution of the paper is to show quantitatively that counterfactual reasoning can be beneficial for learning policies in reinforcement learning, admittedly in a highly idealized but not trivial task. In our opinion, this is an important, novel result, given that humans almost constantly engage in counterfactual reasoning, for which a vague functional role was hypothesised but no learning mechanism has been proposed (see [Roese 97]).
Ultimately, we think our proposed method can contribute to novel methods to the important problem of off-policy learning.

We are currently working on applying the proposed methods to partially observed problems in continuous control, to study if the observed benefits carry over to less idealized settings.

---

### Comment · Area_Chair1 · 2018-12-14
**Interesting contribution, title and introduction to reflect more narrow scope**




This is a clear topic of increasing importance to the community-- combining causality / counterfactual reasoning with sequential decision making. The authors draw their perspective from the view of structural causal modeling and it would also be beneficial to reference the body of literature from more of the potential outcomes framework-- see below for a few of these references in the RL counterfactual/ off policy policy evaluation community. In particular, the proposed approach here is general but only instantiated (in terms of inference algorithms and experiments) for when the initial starting state is unknown in a deterministic POMDP environment, where the dynamics and reward model is known. The authors show that they can use inference over the full trajectory (or some multi-time-step subpart) to get a (often delta function) posterior over the initial starting state, which then allows them to build a more accurate initial state distribution for use in their model simulations than approaches that do not use more than 1 step to do so. This is interesting, but it’s not quite clear where this sort of situation would arise in practice, and the proposed experimental results are limited to one simulated toy domain. This is fine, but the title and introduction seem to suggest a much more general contribution, as opposed to this much more restricted (though interesting) setting of inferring the initial starting state distribution when the dynamics and reward model are known. Therefore I encourage the authors to update their title and introduction to narrow the scope of the proposed contribution.

Mandel, Liu, Levine, Brunskill, Popovic AAMAS 2014.
Thomas and Brunskill ICML 2016
Schulam, Saria. NeurIPS 2017
Guo, Thomas, Brunskill NeurIPS 2017
Parbhoo, Gottesman, Ross, Komorowski, Faisal, Bon, Roth, Doshi-Velez, PloS one 2018
Liu, Gottesman, Raghu, Komorowski, Faisal, Doshi-Velez, Brunskill NeurIPS 2018

---

> ### Author Response · Authors · 2018-12-19
> **Re**
>
> We thank the area chair for pointing out the references, we will add them to our
> manuscript. As stated in the response to the reviewers, we agree that our
> experiments test our algorithm only in the idealized setting of known transition
> and reward kernels and unknown initial state. We will change the wording in the
> introduction to better reflect the scope of our experiments. We maintain
> however, that the underlying idea of inferring scenarios (that can influence all
> transitions) in hindsight from off-policy data, and re-using these for
> counterfactual policy evaluation in principle applies to a wider setting. Given
> the close connection of our proposed algorithm to the GPS algorithm as presented
> by [Levine, Abbeel. 2014], we prefer to keep it's name (CF-GPS) as well as the
> title of the paper as is.

---

### Meta-Review · Area_Chair1 · 2018-12-14
**Interesting idea, scope is much narrower than presentation would suggest**

**Confidence:** 4
**Recommendation:** Accept (Poster)

**Metareview:**

see my comment to the authors below